# An Improved BAQ Encoding and Decoding Method for Improving the Quantized SNR of SAR Raw Data

**DOI:** 10.3390/s18124221

**Published:** 2018-12-01

**Authors:** Wei Ji, Xiaolan Qiu, Xuejiao Wen, Lijia Huang

**Affiliations:** 1Key Laboratory of Technology in Geo-Spatial Information Processing and Application System, Institute of Electronics, Chinese Academy of Sciences, Beijing 100190, China; jiwei16@mails.ucas.ac.cn (W.J.); xjwen@mail.ie.ac.cn (X.W.); lihuang@mail.ie.ac.cn (L.H.); 2Institute of Electronics, Chinese Academy of Sciences, Beijing 100190, China; 3School of Electronic, Electrical and Communication Engineering, University of Chinese Academy of Sciences, Beijing 100049, China

**Keywords:** synthetic aperture radar (SAR), raw data compression, block adaptive quantization (BAQ), anti-saturation, signal-to-noise ratio (SNR)

## Abstract

When the original echo data of SAR are saturated for quantization, the performance of the commonly used block adaptive quantization (BAQ) algorithm will be degraded, which will degrade the imaging quality. This article proposes an improved Llody-Max codec method, which only needs to change the codec look-up table to get better quantization performance when the original echo is saturated. The simulation results show that the proposed method can reduce the quantization power loss, improve the echo signal-to-noise ratio (SNR), and reduce the influence of quantization saturation on the scattering mechanism of polarized SAR data, which have good practical application value.

## 1. Introduction

Synthetic aperture radar (SAR) system technology has been developing towards high resolution, wide swaths, high revisit frequency, multi-frequency, multi-polarization, and so on. The application of SAR has turned from qualitative applications to quantitative applications. Therefore, the improvement of SAR quantitative accuracy has become one of the key aspects in development. The quality of SAR raw echo data is a prerequisite for good SAR image quality. Due to a large amount of SAR raw echo data and the limitation of transmission bandwidth, data rate, and so on, the SAR raw data are usually compressed [1,2,3]. Therefore, quantization and compression will affect the quality of echo and further the SAR images.

There have been quite a lot of studies on SAR raw data compression algorithms [4,5,6]. Considering the complexity of the algorithm, compression time, hardware conditions, and other factors, the block adaptive quantization (BAQ) algorithm is mainly adopted in SAR satellites for its simplicity and efficiency. For example, it has been used in the American Magellan SAR [7], ESA’s Sentinel-1 [8], Canada’s Radarsat-2 [9], Germany’s TanDEM-X [10], Chinese GF-3 SAR [11], and so on. The BAQ algorithm can adaptively quantify the echo with variant power intensity, and get the best quantization SNR for the Gauss distributed echo. However, when the echo is saturated the performance of BAQ will be degraded.

To enhance the performance, and also to reduce the data rate, many improved BAQ methods have been proposed. For example, the flexible dynamic block adaptive quantizer algorithm used by Sentinel-1 is proposed in the literature [8], which is a data compression algorithm that automatically selects a BAQ quantizer out of a set of five quantizers, based on an estimation of the local SNR estimated blocks on the acquired data. In these two methods, the overall performances are improved under limited data rate, but the performance of the saturated data cannot be improved. The literature [10] describes a novel azimuth-switched quantization (ASQ) technique, which provides the capability of synthesizing fractional quantization rates without impacting the complexity and computational load of the quantization scheme. The method has been carried out in the frame of the TanDEM-X mission, and it is shown that performance and resources can be dynamically scaled with a very fine discretization. Zhao et al., in the literature [12], propose a method for selecting a quantization strategy based on the judgment of echo data saturation. In the case of low saturation, the traditional BAQ compression [7] is used, and in the case of high saturation, the saturation data is coded and decoded separately. This method can get better quantization SNR for the saturated data, but it has to increase the saturation judgment, which will increase the amount of computation on the system. Qiu et al. propose a method in the literature [13], which obtains the variance of the input signal by calculating the quantization output power, and then dynamically changes the quantized boundary code value. So, it can improve the performance when the data is saturated, but the improvement is not as significant because it is limited by the coding system. Qi et al. propose a quantization method, based on the optimal nonlinear scalar quantizer and a power compensation decoder, in the literature [14]. The method is based on the relationship between signal saturation, uniform quantified peaks, standard deviation (before and after) uniform quantization, and signal amplitude mean, and they designed a new look-up table and a power compensator. Our studies on this method found that the encoding and decoding methods can be further optimized, and the quantization performance can be further improved.

This article proposes an improved BAQ encoding and decoding method for improving the signal-to-noise ratio of SAR raw data. Through the analysis of uniform quantization and coding processes, this article redesigns the standard deviation and the amplitude mean value look-up table of the uniform quantized signal. The decoding method is redesigned according to the principle of minimum-quantization noise loss. This method significantly improves compression performance, compared with the traditional BAQ compression algorithm [7] and other methods [12,13,14]. This method does not need to make any changes to the quantization hardware system, and only needs to replace the look-up table. There is no increment in the complexity and computation of the algorithm. The rest of the article is arranged as follows: In Section 2, the improved BAQ compression algorithm is described. Section 3 gives the performance analysis results, through simulation and practical experiments to verify the effectiveness of the method. Section 4 concludes the article.

## 2. The Improved BAQ Encoding and Decoding Method

The traditional BAQ method first divides the input signal into blocks, then uniformly quantizes, normalizes, and non-uniformly quantizes the data for each block, and then decodes it to get the output signal, as shown in Figure 1. When N bit non-uniform quantization was performed on the original SAR echo, the quantitative dynamic range was $\left[-2^{N-1},2^{N-1}\right]$, and all values larger than $\left(2^{N-1}-1\right)$ and smaller than $\left(-2^{N-1}+1\right)$ were quantified as $\left(2^{N-1}-0.5\right)$ and $\left(-2^{N-1}+0.5\right)$, respectively. So, when the absolute value of the signal is greater than $2^{N-1}$, quantization saturation occurs. Quantization saturation results in a truncation effect, leading to the loss of effective quantization intervals, which ultimately leads to power loss and affects the quality of the image.

As the Lloyd-Max quantizer (adopted by the traditional BAQ method) requires the quantization signal to conform to the standard Gaussian distribution, it is necessary to normalize the signal after uniform quantization. In the actual system, in order to reduce the amount of computation, a pre-designed mean standard deviation look-up table is adopted, to find the standard deviation according to the mean value and then realize normalization. Quantization saturation can lead to a mismatch of the mapping relationship between the mean value and standard deviation of signal amplitude in traditional BAQ methods, as shown in Figure 2. If the original variance is used for normalization, although the unsaturated part still satisfies the standard Gaussian distribution, the absence of effective quantization interval will be caused, due to the truncation effect. Therefore, the standard deviation adopted by the traditional method can no longer be used in the case of quantitative saturation, but should be solved again according to the truncated signal [14], as shown by the red curve in Figure 2.

In the case of non-uniform quantization, taking 8:3 BAQ as an example, when the signal after 8 bit uniform quantization is not saturated, the normalized peak data will fall in the eighth quantization interval, so that all quantization intervals are effective, as shown in Figure 3a. When the signal after 8 bit uniform quantization is saturated, the data will be truncated. If normalized by the standard deviation in the traditional method, it will cause the normalized peak data to fall into a certain quantization interval, resulting in an invalid part of the quantization interval. As shown in Figure 3b, the peak value falls into the seventh interval after normalization, resulting in the invalidation of the first and eighth quantization intervals. Then, if the original method is also adopted for encoding and decoding, it will inevitably lead to large quantization noise and cannot achieve a good quantization effect. Therefore, it is necessary to recalculate the standard deviation for normalization, as well as the threshold and quantization level of each quantization interval.

In order to achieve a better compression effect, this article makes two improvements: First, we redesign the mean standard deviation look-up table for normalization; Second, we improve the threshold and quantization level of the individual quantization intervals for the Lloyd-Max quantizer, which is the core of BAQ.

The improved method for the mean standard deviation look-up table is as follows: In practice, satellite coding is based on the average, to find the corresponding standard deviation and quantization threshold of the table for normalization. For convenience, this article improves the original normalization method. The signals without saturation (after uniform quantization) are still normalized with the original standard deviation, and the signals with saturation (after uniform quantization) are normalized with the standard deviation described by the red curve in Figure 2. When decoding, the unsaturated case can be decoded according to the original mode, and the saturated case can be decoded, according to the corresponding table when encoding.

The improved method for the Lloyd-Max quantizer is as follows: The probability distribution of the signal after uniform quantization can be expressed as:
(1)$$\left\{ \begin{array}{l} f_{1}(x)=(\int_{-M}^{-\infty }f_{2}(\mu )d\mu )\cdot \delta (x);x\leq -M\\ f_{2}\left(x\right)=\frac{1}{\sqrt{2\pi }\sigma }\exp\left\{-\frac{x^{2}}{2\sigma^{2}}\right\};-M<x<M\\ f_{3}(x)=(\int_{M}^{\infty }f_{2}(\mu )d\mu )\cdot \delta (x);x\geq M \end{array}\right.,$$
where σ is the standard deviation of the signal after uniform quantization, *M* is the value at the saturation threshold, and δ(x) is the impulse function, which represents the probability value at the “truncated” point, and is equal to the probability that the unsaturated signal goes from truncation to positive infinity. According to the definition, the distortion of the quantized signal is:
(2)$$D=\sum_{i=1}^{N}\int_{x_{i}}^{x_{i+1}}{\left(x-y_{i}\right)}^{2}f(x)dx,$$
where $x_{i}$ is the threshold level, $y_{i}$ is the quantization level, and $x_{N+1}$ is defined as positive infinity. By calculating the partial derivatives of $x_{i}$ and $y_{i}$ in the Equation (2), the minimum value of the quantized signal distortion can be obtained.
(3)$$x_{i}=\frac{y_{i}+y_{i-1}}{2},i=2,\cdots ,N$$
(4)$$\int_{x_{i}}^{x_{i+1}}(x-y_{i})f(x)dx=0,i=1,\cdots ,N$$

To further consider the impact of saturation, we need to re-solve the required $x_{i}$ to get the minimum quantization loss ${y_{i}}^{\prime }$, corresponding to $x_{i}$. Equation (4) is shifted and sorted:
(5)$${y_{i}}^{\prime }=\frac{\int_{x_{i}}^{x_{i+1}}xf(x)dx}{\int_{x_{i}}^{x_{i+1}}f(x)dx},i=1,\cdots ,N.$$

According to the above recursion, we can get $x_{i}$ and ${y_{i}}^{\prime }$ with the smallest quantization loss in the saturation case. Taking the 8:3 BAQ as an example, the specific solution steps for the threshold level $x_{i}$ and quantization level ${y_{i}}^{\prime }$ of the improved Lloyd-Max quantizer are as follows:
(1)For the 8:3 BAQ, N equals 4 and we need to set $x_{1}$ and $y_{1}$, where $x_{1}$ is zero and the range of $y_{1}$ is known to be between 0.1 and 0.3, based on prior knowledge. We perform iterative calculations according to Equations (3) and (4) to obtain $x_{2}$, $x_{3}$, $x_{4}$, $y_{2}$, $y_{3}$, and $y_{4}$, corresponding to different values of $y_{1}$.(2)Calculate the corresponding ${y_{1}}^{\prime }$, ${y_{2}}^{\prime }$, ${y_{3}}^{\prime }$, and ${y_{4}}^{\prime }$, according to Formula (5), for each group $x_{1}$, $x_{2}$, $x_{3}$, and $x_{4}$.(3)According to Formula (2), the quantization distortion is calculated for each group $x_{1}$, $x_{2}$, $x_{3}$, $x_{4}$, ${y_{1}}^{\prime }$, ${y_{2}}^{\prime }$, ${y_{3}}^{\prime }$, and ${y_{4}}^{\prime }$. Then, select the set of solutions corresponding to the smallest quantization distortion: Namely, the optimal threshold level and the optimal quantization level.(4)For the different input signals, repeat the above steps to obtain different optimal threshold levels and optimal quantization levels, under different f(x).

The flowchart is as follows (Figure 4):

The overall method of this article is as follows:
(1)Divide a pulse of the SAR raw echo into blocks. Take M sampling points of one block (M is generally taken as 1024 or 512).(2)The SAR raw echo is sampled and N bit uniformly quantized, where N is generally 8;(3)The average absolute amplitude of the I and Q signals in each block is calculated, respectively, denoted as A. According to the value of A, find the corresponding standard deviation in the new standard deviation look-up table for normalization;(4)For the normalized data, non-uniform quantization coding is performed, according to the optimal threshold level corresponding to each standard deviation obtained by the method in this article;(5)During decoding, according to the corresponding standard deviation, the corresponding decoding method is adopted, so as to achieve the purpose of data recovery.

The flowchart is as follows (Figure 5):

## 3. Analysis and Results

In order to prove the effectiveness of this method, this article compares the results of the proposed method and the existing methods, from the aspects of echo quantized signal-to-noise ratio, quantization power loss, and polarization scattering mechanism retention ability.

### 3.1. Quantization Loss and Quantization SNR

First, we simulate Gaussian-distributed data at different input powers and then uniformly quantize the data. Next, we average the uniformly quantized data and find the look-up table (described in Section 2), to get the standard deviation corresponding to the average value to normalize. Then, we encode and decode the normalized data as described in Section 2. Finally, we find and compare the SNR and quantized power loss of the processed data. Compared with the original BAQ [7] and several existing anti-saturation methods [13,14], the results are shown in Figure 6 and Figure 7.

It can be seen from Figure 6 that, with the increase of input power, the quantitative power loss of the method in this article is always minimal, and the advantage of this method is more obvious when the input power is 40–60 db. From Figure 7a, we intuitively see that, after the echo data saturation in this method, the quantized SNR is optimal, which is 3–4 dB higher than the average of the original method [7]. Compared with Qiu’s method [14], it has an average of 2–3 dB improvement. Compared with Qi’s method, it has an average of 1–2 dB improvement. When the SNR is greater than 12 dB, the input power range of this method is 8–41.7 dB, and 8–37.1 dB for the other methods, nearly 4 dB more for this method. In order to more intuitively embody the anti-saturation function of the proposed method, the data saturation and quantization signal-to-noise ratio are statistically analyzed. As shown in Figure 7b, we can see that the SNR of the proposed method is greater than or equal to the existing anti-saturation method at different saturations. Saturation refers to the ratio of the number of data points exceeding the range of $\left[-2^{N-1},2^{N-1}\right]$ to the total, in the process of uniform quantization.

From Figure 8, we obtain the effective quantization interval, threshold, and quantization level of the four methods when the input power is 45 dB. There are only four effective quantization intervals for the original method and Qiu’s method, six effective quantization intervals for Qi’s method, and eight effective quantization intervals for our method.

In summary, according to the simulation results, this result is consistent with the previous analysis. This method has better anti-saturation performance.

### 3.2. Influence on Polarization Scattering Mechanism

SAR raw data compression can reduce on-board downlink data rate effectively. However, SAR raw data compression induces distortions on polarimetric information of quad-polarimetric SAR data. In order to analyze the influence of different quantization compression methods on the characteristics of polarimetric SAR, we selected the unsaturated polarization data of GF-3 in the San Francisco area and added quadratic phase to simulate the echo. We controlled the saturation of the data by controlling the input power, and then made quantization compression with the traditional BAQ method, Qi’s method, Qiu’s method, and the method in this article, and analyzed their polarization characteristics. The images after the quantization compression processing and power loss difference are shown in Figure 9 and Figure 10.

It can be seen intuitively, in Figure 10, that the power loss of the Qi’s quantization compression method is large, and the image relative brightness appears obviously reduced. By adopting Qiu’s quantization compression method, the power loss is reduced and the visual effect of the image is relatively improved. With the quantization compression method in this article, the power loss is minimized and the visual effect of the image is closer to the original image. It can be seen, in Figure 10, that the power gain of our method also has relatively good results.

This article then compares the target scattering characteristics of three quantitative compression methods, including sea, forest, and building area, as shown in Figure 9a. In the H-alpha plane, alpha represents the scattering angle, and H represents the scattering entropy. We can classify the target based on its position in the H-alpha plane. The H-alpha plane graph and the interval histogram of the three methods are shown in Figure 11, Figure 12 and Figure 13.

From the scattering characteristics of the target, the scattering characteristics of the proposed method are superior to the other two methods in the sea, forest, and building areas, and closer to the target scattering characteristics of the original image. In summary, this method is superior to the other two methods, in both visual effect and target scattering characteristics, for saturated data processing.

## 4. Conclusions

This article proposes an anti-saturation coding and decoding method for BAQ, commonly used in satellite-borne SAR. The simulation data and actual data show that the performance of this method is better than that of the traditional and existing anti-saturation methods, and this method does not need to modify the hardware system on the satellite, which can play a good role in correcting the saturation echo. It reduces the quantization power loss, improves the image SNR, and has important engineering application value. 

## Figures and Tables

**Figure 1 sensors-18-04221-f001:**
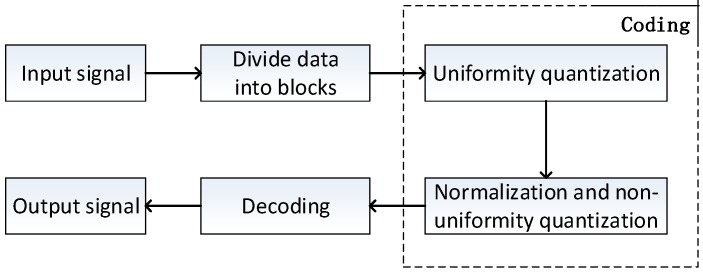
The flowchart of traditional BAQ method.

**Figure 2 sensors-18-04221-f002:**
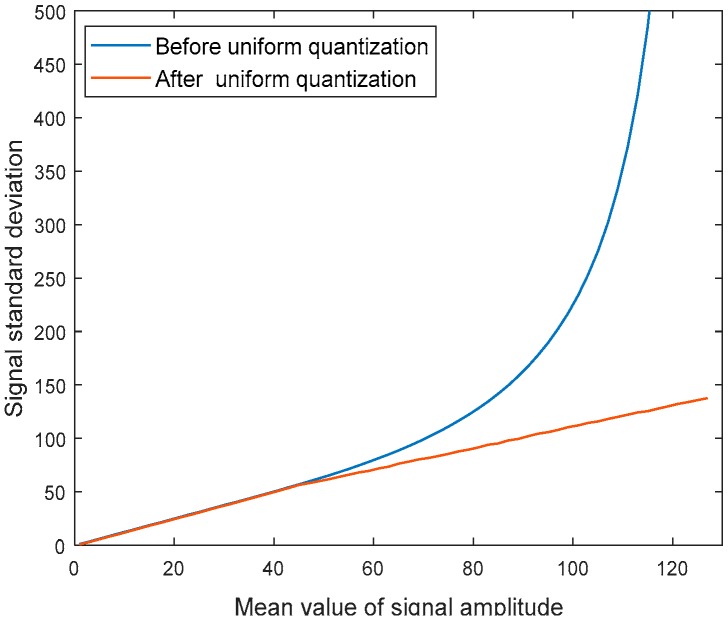
The relationship between the signal standard deviation and mean value of signal amplitude, before and after uniform quantization.

**Figure 3 sensors-18-04221-f003:**
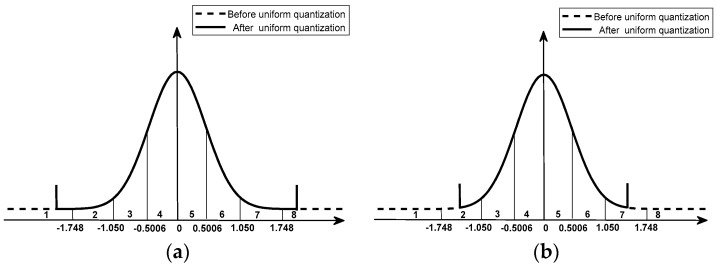
Quantization interval distribution for 8:3 BAQ coding: (**a**) eight quantization intervals are valid; (**b**) six quantization intervals are valid.

**Figure 4 sensors-18-04221-f004:**
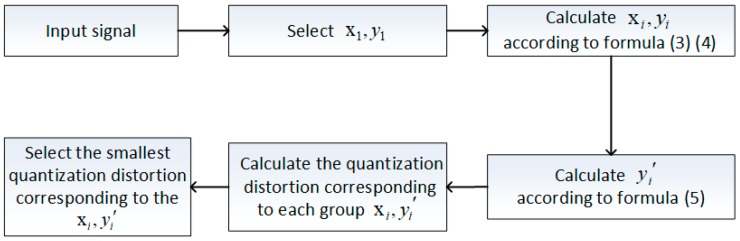
Flowchart for calculating the threshold and quantization level.

**Figure 5 sensors-18-04221-f005:**
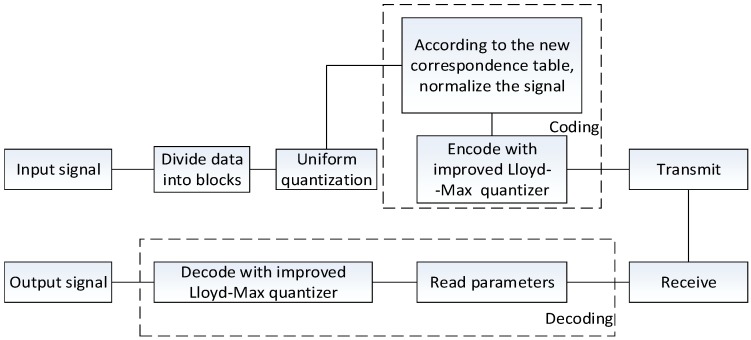
Flowchart of the overall method.

**Figure 6 sensors-18-04221-f006:**
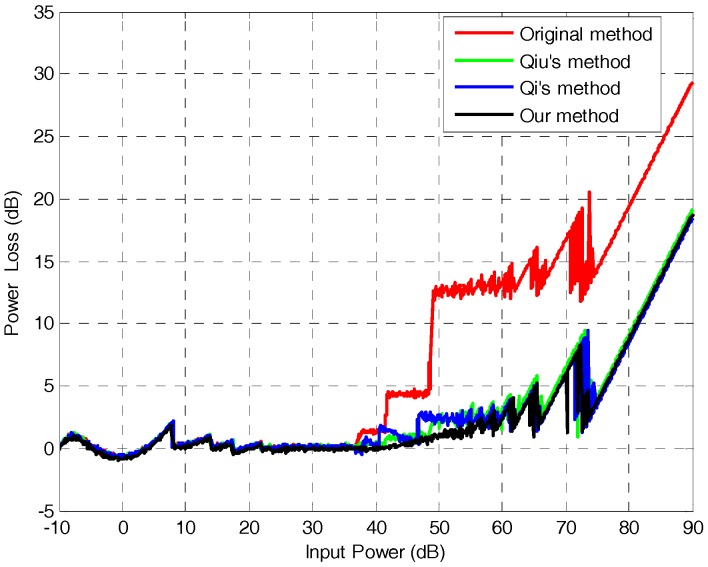
Comparison of input power and quantization power loss.

**Figure 7 sensors-18-04221-f007:**
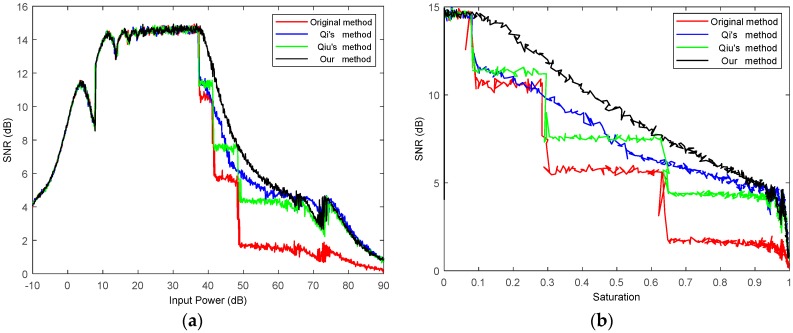
(**a**) Comparison of input power and SNR; (**b**) comparison of saturation and SNR.

**Figure 8 sensors-18-04221-f008:**
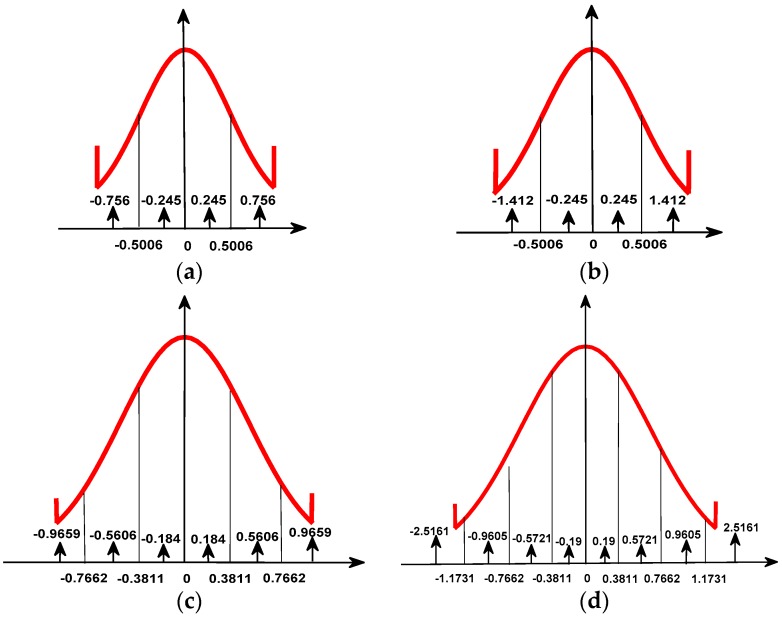
Effective quantization interval, threshold, and quantization level when the input power is 45 dB: (**a**) Original method; (**b**) Qiu’s method; (**c**) Qi’s method; (**d**) our method.

**Figure 9 sensors-18-04221-f009:**
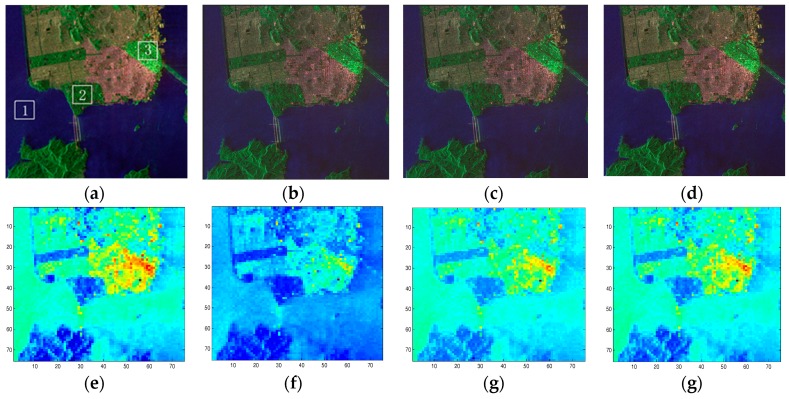
Imaging effect of three 8:3 BAQ quantization compression methods: (**a**,**e**) Original image; (**b**,**f**) Qi’s method; (**c**,**g**) Qiu’s method; (**d**,**h**) our method.

**Figure 10 sensors-18-04221-f010:**
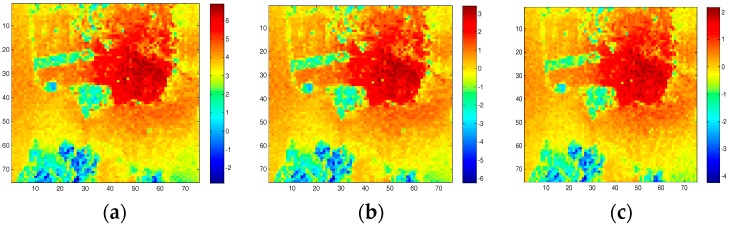
Power loss of three 8:3 BAQ quantization compression methods: (**a**) Qi’s method; (**b**) Qiu’s method; (**c**) our method.

**Figure 11 sensors-18-04221-f011:**
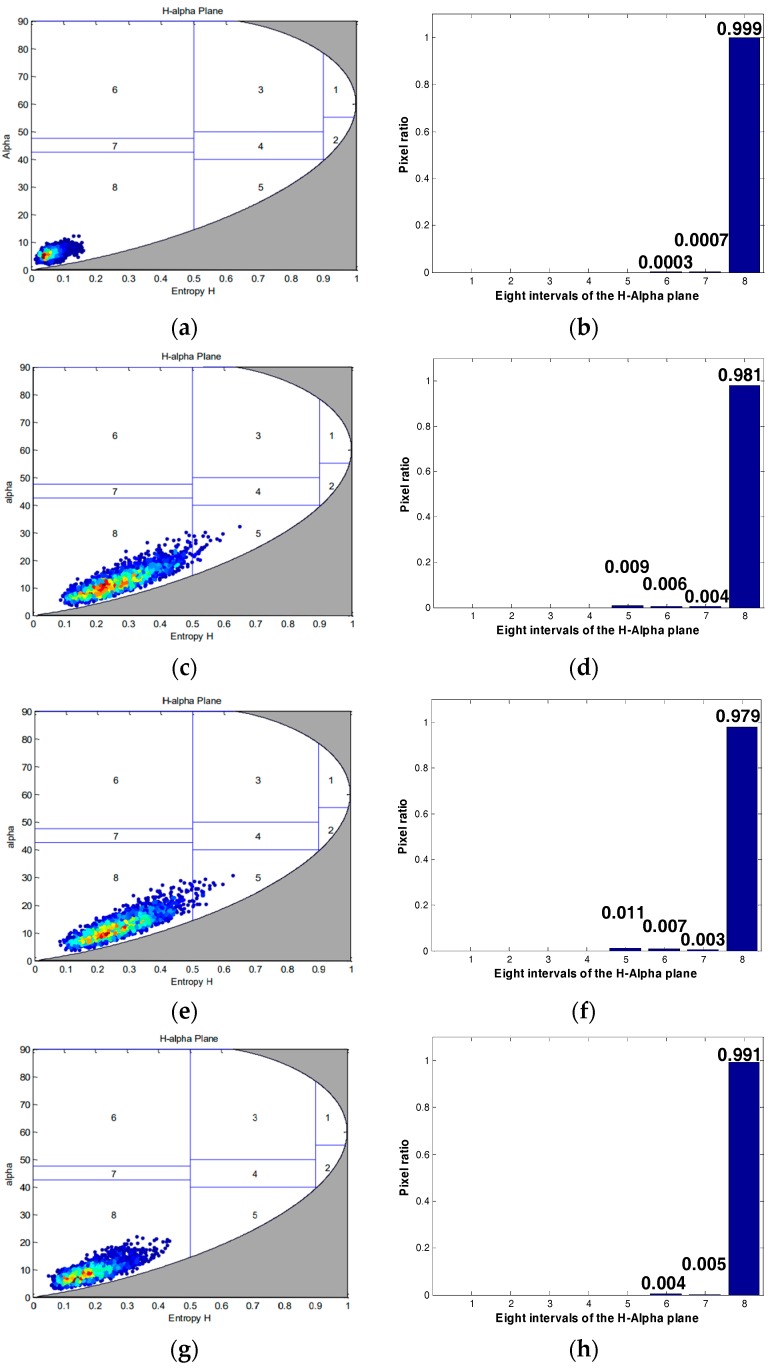
Sea area scattering characteristics under three quantitative compression methods: (**a**,**e**) Original image; (**b**,**f**) Qi’s method; (**c**,**g**) Qiu’s method; (**d**,**h**) our method.

**Figure 12 sensors-18-04221-f012:**
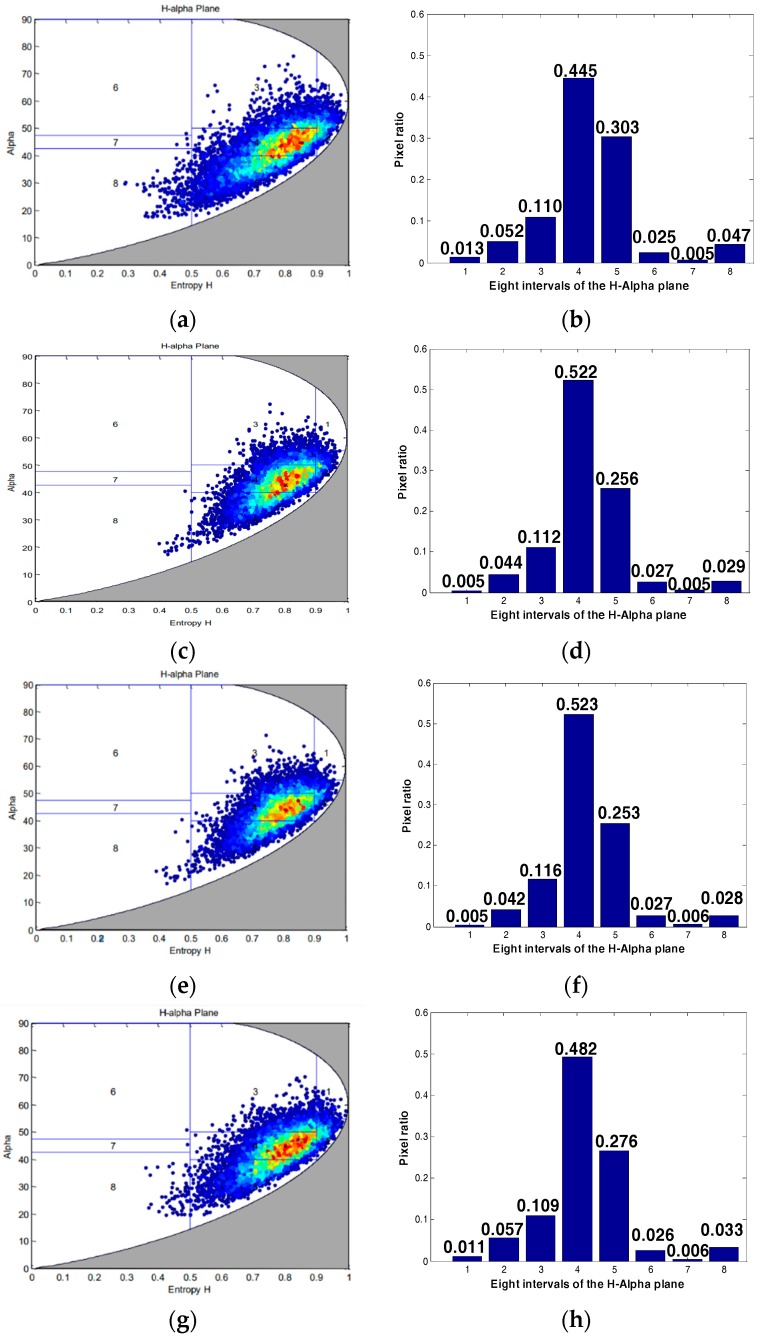
Forest scattering characteristics under three quantitative compression methods: (**a**,**e**) Original image; (**b**,**f**) Qi’s method; (**c**,**g**) Qiu’s method; (**d**,**h**) our method.

**Figure 13 sensors-18-04221-f013:**
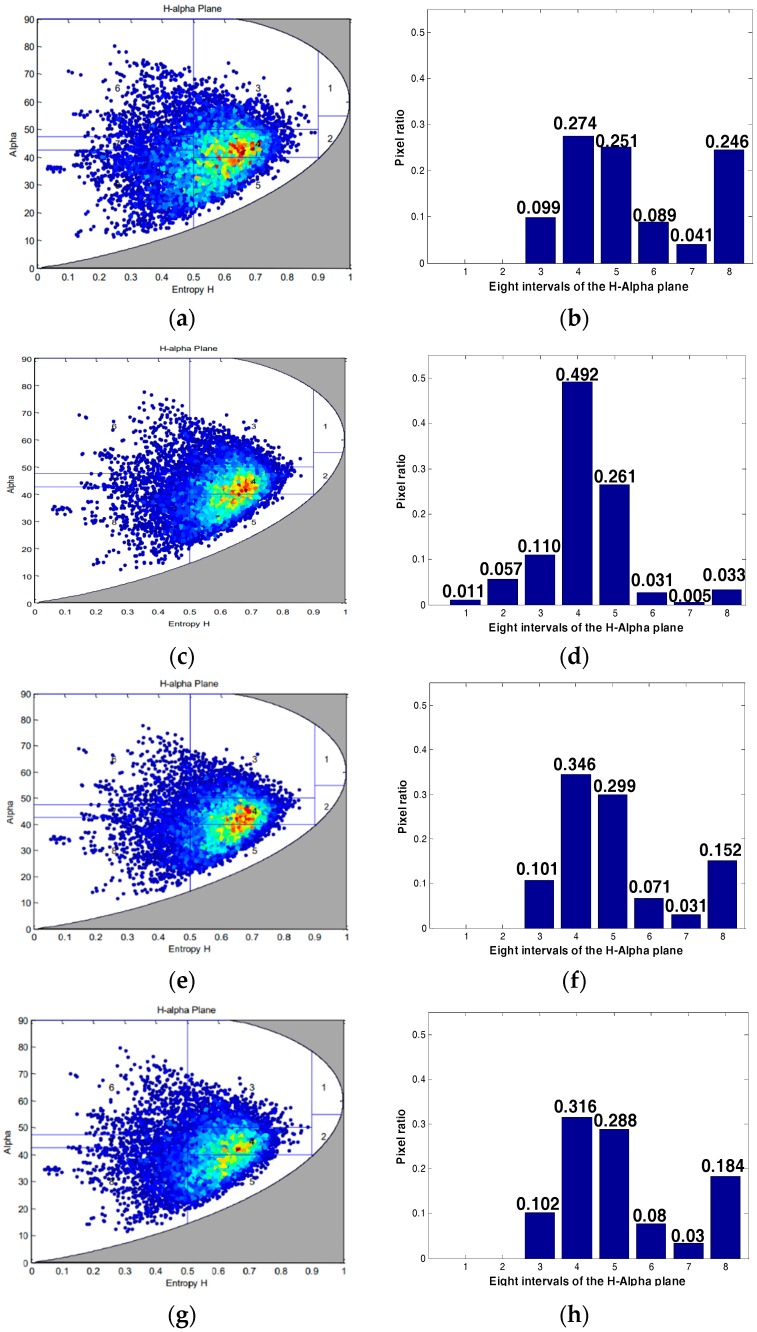
Building area scattering characteristics under three quantitative compression methods: (**a**,**e**) Original image; (**b**,**f**) Qi’s method; (**c**,**g**) Qiu’s method; (**d**,**h**) our method.

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
