# Peer review of "An Improved BAQ Encoding and Decoding Method for Improving the Quantized SNR of SAR Raw Data"

_sensors, 2018, doi:10.3390/s18124221_

Reviewer 1 Report

This paper proposes a novel quantization method for Synthetic Aperture Radar. The proposal is interesting however; various modifications are required prior to be published.

1. In L. 32, the authors wrote “There have been quite a lot of studies on SAR raw data compression algorithms [1-4].” However, some articles are proposal of the method while the others are review and report that they applied a compression method to a specific mission. Please cite the references more effectively. For example, cite the review articles firstly and then, cite the missions which applied the compression methods.
2. Please describe the equations or show the reference for drawing Figure 2.
3. In order to lead the improvements in L. 114-116, please describe the logic that the proposed method can solve the problems in L. 102-111. Provide a traditional, i.e., conventional, BAQ equations and show a problem. Then leading the proposed method becomes more sounding.
4. In Section 3, explain the experimental conditions precisely. It is impossible to understand how the authors derive those graphs. For the simulation results, it is necessary to write sufficient amount of information, i.e., all readers must be able to draw the same results only from the information in the article.
5. In figure 6 and 7, there is no graph which represents [13]. If "Original" in Figure 6 and 7 represent [13], replace "Original" to "Zhao's method" as same as the others.
6. The authors used GF-3 data in Section 3.2. However, according to Section 1, GF-3 applies quantization method. Therefore, the raw data must be quantized naturally. In other words, some strong signals have been saturated naturally and thus, the results do not reflect the fair comparison. Please briefly note and discuss about it.
7. Figure 9 is too small. At least, in Fig. 9(a), the numbers in the rectangles cannot be read. Probably those numbers correspond to the H-alpha graph in Fig. 10-12 but it is not clearly written. The power scale of Fig. 9(f-h) does not match. At the same time, the authors discuss only for power loss but not for power gain, which are also an important drawback of quantization.
8. There are massive grammatical errors. The followings are examples.
8-1. In L. 119-120: “For reasons of convenience, this paper is based on the original normalization method was improved.”
8-2. In L. 171-172: “We simulates the echo signals at different input powers, and then quantifies them according to the methods described in this paper, and calculates the quantized SNR.”
8-3. In L. 204: “SAR raw data compression can reduce on-board downlink rata effectively.” rata -> data?

Author Response

Dear editor,

According to the editor and reviewers’ comments, we have made extensive modifications to our manuscript. In this revised version, changes to our manuscript were all highlighted within the document by an underline. Based on the reviewers’ comments, we also attached a point-by-point letter to each reviewer. We have made extensive revisions to our previous draft. The detailed point-by-point responses are listed below.

 Response to reviewer :

(1) In L. 32, the authors wrote “There have been quite a lot of studies on SAR raw data compression algorithms [1-4].” However, some articles are proposal of the method while the others are review and report that they applied a compression method to a specific mission. Please cite the references more effectively. For example, cite the review articles firstly and then, cite the missions which applied the compression methods.
Thank you for your suggestion, I have already modified these issues in the article. In Section 1, I cited the review articles firstly. Next I cited the articles about data compression algorithms. And then I cited the missions which applied the compression methods.

(2) Please describe the equations or show the reference for drawing Figure 2.
Thank you for your suggestion, I have already cited the literature [14] above Figure 2. This figure is simulated according to the equations (2) in the literature [14].

(3) In order to lead the improvements in L. 114-116, please describe the logic that the proposed method can solve the problems in L. 102-111. Provide a traditional, i.e., conventional, BAQ equations and show a problem. Then leading the proposed method becomes more sounding.

Thank you for your suggestion, I have already modified these issues in the article. The traditional method is not perfect for data saturation. When the data is saturated, if it is also normalized by the standard deviation used in the traditional method, there will be a situation similar to the effective quantization interval missing in Figure.3(b). Therefore, the normalization based on the standard deviation deduced from the literature [14] can accurately and effectively solve the quantitative intervals missing problem. However, the design of thresholds and quantization levels for non-uniform quantization in [14] is not complete. So we redesigned the threshold level and quantization level of non-uniform quantization, which can further optimize the compression effect, as described in the flow in Figure 4.

(4) In Section 3, explain the experimental conditions precisely. It is impossible to understand how the authors derive those graphs. For the simulation results, it is necessary to write sufficient amount of information, i.e., all readers must be able to draw the same results only from the information in the article.

Thanks for your suggestion, I have added description in Section 3, both in 3.1 and 3.2.

In Section 3.1, we analyzed the effect of quantization compression on the signal itself. First, we simulate Gaussian-distributed data at different input powers and then uniformly quantize the data. Next, we average the uniformly quantized data, and find the look-up table described in Section 2 to get the standard deviation corresponding to the average value to normalize. Next, we encode and decode the normalized data as described in Section 2. Finally, we find and compare the SNR and quantized power loss of the processed data.
In Section 3.2, we analyzed the effect of quantization compression on the polarization characteristics of SAR images. So here we choose a proper quantified data which has no saturation and have high quantization SNR. Simulation is performed on the basis of this data. We do imaging and get the focused image, and then do the inverse processing to get the float data type raw echo, then control the gain and do the quantization by different methods. The flow of the experiment in Section 3 of this article can be referred to Figure 5.

(5) In figure 6 and 7, there is no graph which represents [13]. If "Original" in Figure 6 and 7 represent [13], replace "Original" to "Zhao's method" as same as the others.

Thanks for your suggestion. Actually, the "Zhao's method" in the literature [13] was simulated during our experiments, and the quantized SNR of the saturated data was improved by 3-4dB. Considering the actual effect, this method is not as good as Qiu’s method and Qi’s method, and the aesthetics of Figure 6 and Figure 7. So it is not shown in the figure.

(6)The authors used GF-3 data in Section 3.2. However, according to Section 1, GF-3 applies quantization method. Therefore, the raw data must be quantized naturally. In other words, some strong signals have been saturated naturally and thus, the results do not reflect the fair comparison. Please briefly note and discuss about it.

YesIt is true that the GF-3 data has been quantified, and there is no way to do experiment with unquantified raw data. So here we choose a proper quantified data which has no saturation and have high quantization SNR. Simulation is performed on the basis of this data. We do imaging and get the focused image, and then do the inverse processing to get the float data type raw echo, then control the gain and do the quantization by different methods. We then do imaging to this different quantization echo to get the images and do the following analysis. We have added description in section 3.2 to explain the simulation.

 (7) Figure 9 is too small. At least, in Fig. 9(a), the numbers in the rectangles cannot be read. Probably those numbers correspond to the H-alpha graph in Fig. 10-12 but it is not clearly written. The power scale of Fig. 9(f-h) does not match. At the same time, the authors discuss only for power loss but not for power gain, which are also an important drawback of quantization.

Thanks for your suggestion, I have replaced Fig. 9(a) with an image of the appropriate size. In addition, we selected three representative areas in the figure, namely water, forest, and building to analyze the scattering characteristics. Figure 1-3 in Fig. 9(a) correspond to Figure 11-13 (Figure 10-12 before modification). In addition, Fig. 9(e) is the power map of the original image. Fig. 9(f)-(h) before the modification are the power loss figures. Since the processed power map is similar to the power map of the original image, the figure before the modification is shown in Fig. 9(f)-(h) is placed with a power loss graph. Since the color of the power maps processed by the three methods are similar to the power map of the original image, so the power loss maps are placed in Fig. 9(f)-(h) (before modification). Since considering that may affect readers' reading, the article made the following changes. We put the same power diagram as Fig. 9(e) in Fig. 9(f)-(h) and add Fig.10(a)-(c) to plot the power loss. So we can compare the power loss and power gain of several methods with the colorbar in Figure 10.

 (8) There are massive grammatical errors. The followings are examples.

8-1. In L. 119-120: “For reasons of convenience, the paper is based on the original normalization method was improved.”

8-2. In L. 171-172: “We simulates the echo signals at different input powers, and then quantifies them according to the methods described in the paper, and calculates the quantized SNR.”

8-3. In L. 204: “SAR raw data compression can reduce on-board downlink rata effectively.” rata -> data?

Thank you for your suggestion, I carefully checked the entire article and made a careful revision. The modified part has been underlined.

Thank you for the nice comments again, and we hope that you will be satisfied with our responses.

 Best regards,
Wei Ji
E-mail: [email protected]
Corresponding author:
Name: Xiaolan Qiu
E-mail: [email protected]

Reviewer 2 Report

This paper presents an improved BAQ scheme for SAR systems. The reviewer has some suggestions.

For Fig. 3 and the procedure regarding the uniform quantizer on page 4, the relation between the threshold or range of the uniform quantizer and the input variance should be defined so that it can be realized the saturation effect when an input power is given. This relation is also important to realize Fig.8.
Please slightly introduce the Qi's method and Qui's method.
What is \delta(x) in (1)? Please define it.
On page 7, please define the meaning of the value [0-1] for saturation in Fig. 7 (b).
The color bar of Fig. 9(e) is strange compared to Fig. 9(f)-(h). Please make the color bar setting consistent among these figures so that we can check the figures from the colors.
Please define alpha of Fig. 10-12. Please define the meaning of X-axis of Fig. 10-12(e)-(h).

Author Response

Dear editor,

According to the editor and reviewers’ comments, we have made extensive modifications to our manuscript. In this revised version, changes to our manuscript were all highlighted within the document by an underline. Based on the reviewers’ comments, we also attached a point-by-point letter to each reviewer. We have made extensive revisions to our previous draft. The detailed point-by-point responses are listed below.

 Response to reviewer :

(1) For Fig. 3 and the procedure regarding the uniform quantizer on page 4, the relation between the threshold or range of the uniform quantizer and the input variance should be defined so that it can be realized the saturation effect when an input power is given. This relation is also important to realize Fig. 8.

Thanks for your suggestion, I have added the corresponding threshold in Fig.8. In addition, we can know from the curve of Fig.7(a) that the optimal input power is 20-37dB, and it can be also known that more than 37dB will have quantitative saturation for 8:3BAQ.

 (2) Please slightly introduce the Qi's method and Qiu's method.

Thank you for your suggestion, in order to make it easier for readers to read, I made a slight change in the article, adding the author's name to literature 14 and literature 15 in the first section.

 (3) What is \delta(x) in (1)? Please define it.

\delta(x) is an impulse function, indicating that the curve is "truncated" here, and the probability of "truncation" is equal to the probability that there is no truncated signal to positive infinity. The proof process is detailed in the  literature [14]. Thanks for your suggestion, I have already defined it in the article.

 (4) On page 7, please define the meaning of the value [0-1] for saturation in Fig. 7 (b).

 [0-1] refers to saturation, and saturation refers to the ratio of the number of data exceeding 2^N-1 or -2^N-1 to the total in the process of uniform quantization. Thank you for your suggestion, I have already defined it in the article.

 (5) The color bar of Fig. 9(e) is strange compared to Fig. 9(f)-(h). Please make the color bar setting consistent among these figures so that we can check the figures from the colors.

Fig. 9(e) is the power map of the original image. Fig. 9(f)-(h) before the modification are the power loss figures. Since the processed power map is similar to the power map of the original image, the figure before the modification is shown in Fig. 9(f)-(h) is placed with a power loss graph. Since the color of the power maps processed by the three methods is similar to the power map of the original image, so the power loss maps are placed in Fig. 9(f)-(h) (before modification). Since considering that may affect readers' reading, the article made the following changes. We put the same power diagram as Fig. 9(e) in Fig. 9(f)-(h) and add Fig.10(a)-(c) to plot the power loss. So we can compare the power loss and power gain of several methods with the colorbar in Figure 10.

 (6)Please define alpha of Fig. 10-12. Please define the meaning of X-axis of Fig. 10-12(e)-(h).

In Fig. 10-12 (the modified Fig. 11-13), alpha refers to the scattering angle and H refers to the scattering entropy. Fig. 10-12(e)-(h) (the modified Fig. 11-13(e)-(h)) represents the X coordinate of Fig. 10-12(a)-(d) (the modified Fig 11-13(a)-(d)) Probability that the pixel points are distributed over eight scattering intervals of the H-alpha plane. Thank you for your suggestion, I have already defined it in the article.

 Thank you for the nice comments again, and we hope that you will be satisfied with our responses.

 Best regards,

Wei Ji
E-mail: [email protected]
Corresponding author:
Name: Xiaolan Qiu
E-mail: [email protected]

Round  2

Reviewer 1 Report

The revised paper seems fine to be published.

Please show Fig. 9 and 10 with the same scale.

Author Response

Thank you for the nice comments, I have already modified these issues in the article.And we hope that you will be satisfied with our responses.

Reviewer 2 Report

The authors have addressed the comments adequately.

Author Response

Thank you for the nice comments,and we hope that you will be satisfied with our responses.